# *Clostridioides difficile* Infection: Use of Inflammatory Biomarkers and Hemogram-Derived Ratios to Predict Mortality Risk in Hospitalized Patients

**DOI:** 10.3390/antibiotics13080769

**Published:** 2024-08-15

**Authors:** Giuseppe Guido Maria Scarlata, Angela Quirino, Carmen Costache, Dan Alexandru Toc, Nadia Marascio, Marta Pantanella, Daniel Corneliu Leucuta, Abdulrahman Ismaiel, Dan Lucian Dumitrascu, Ludovico Abenavoli

**Affiliations:** 1Department of Health Sciences, University of Catanzaro “Magna Graecia”, 88100 Catanzaro, Italy; giuseppeguidomaria.scarlata@unicz.it (G.G.M.S.); l.abenavoli@unicz.it (L.A.); 2Unit of Clinical Microbiology, Department of Health Sciences, University of Catanzaro “Magna Graecia”, 88100 Catanzaro, Italynmarascio@unicz.it (N.M.);; 3Emergency Clinical County Hospital, 400000 Cluj-Napoca, Romania; 4Department of Microbiology, “Iuliu Hațieganu” University of Medicine and Pharmacy, 400012 Cluj-Napoca, Romania; 5Department of Medical Informatics and Biostatistics, “Iuliu Hațieganu” University of Medicine and Pharmacy, 400012 Cluj-Napoca, Romania; 62nd Department of Internal Medicine, “Iuliu Hațieganu” University of Medicine and Pharmacy, 400006 Cluj-Napoca, Romania; ismaiel.abdulrahman@umfcluj.ro (A.I.);

**Keywords:** *Clostridioides difficile*, infection control, antimicrobial stewardship, personalized medicine, COVID-19, predictors

## Abstract

Background: *Clostridioides difficile* infection (CDI) is a significant cause of mortality, especially in healthcare environments. Reliable biomarkers that can accurately predict mortality in CDI patients are yet to be evaluated. Our study aims to evaluate the accuracy of several inflammatory biomarkers and hemogram-derived ratios in predicting mortality in CDI patients, such as the neutrophil-to-lymphocyte ratio (NLR), the systemic immune-inflammation index (SII), the platelet-to-neutrophil ratio (PNR), the derived neutrophil-to-lymphocyte ratio (dNLR), C-reactive protein (CRP), the platelet-to-lymphocyte ratio (PLR), and procalcitonin (PCT). Results: NLR showed a sensitivity of 72.5% and a specificity of 58.42% with an area under curve (AUC) = 0.652. SII had a sensitivity of 77.5%, a specificity of 54.74%, and an AUC = 0.64. PNR, neutrophils, dNLR, and lymphocytes had lower AUCs which ranged from 0.595 to 0.616, with varied sensitivity and specificity. CRP, leukocytes, and platelets showed modest predictive values with AUCs below 0.6. PCT had a sensitivity of 100%, a low specificity of 7.41%, and an AUC = 0.528. Methods: We conducted a retrospective analysis of CDI patients from two different hospital settings in Italy and Romania during the COVID-19 pandemic, from 1 January 2020 to 5 May 2023. Statistical analyses included *t*-tests, Wilcoxon rank-sum tests, χ2 tests, and multivariate logistic regression to identify predictors of mortality. ROC analysis assessed the accuracy of biomarkers and hemogram-derived ratios. A *p* value < 0.05 was considered significant. Conclusions: Neutrophils, dNLR, NLR, SII, and PNR are valuable biomarkers for predicting mortality in CDI patients. Understanding these predictors can improve risk stratification and clinical outcomes for CDI patients.

## 1. Introduction

*Clostridioides difficile* (*C. difficile*), previously known as *Clostridium difficile*, is a Gram-positive, spore-forming anaerobic bacterium. It is an opportunistic pathogen primarily affecting the gastrointestinal tract, where it can cause significant morbidity and mortality [1]. *C. difficile* is characterized by its ability to produce two major toxins, TcdA and TcdB, which disrupt the epithelial lining of the intestine, leading to inflammation and diarrhea [2]. This event is crucial in the pathogenesis of *C. difficile* infection (CDI), which contributes to a wide range of clinical manifestations, from mild diarrhea to severe pseudomembranous colitis [3]. Additionally, the spores produced by the pathogen are highly resistant to environmental stress and can persist in healthcare environments, facilitating transmission and infection [2]. Their persistence underscores the challenge of controlling CDI outbreaks in healthcare settings. CDI has emerged as a major healthcare-associated infection worldwide, with increasing incidence over the years (the time period from 2001 to 2012 showed a 46% increase) [4]. The epidemiology of CDI shows a higher prevalence in older adults, particularly those with prolonged hospital stays or those residing in long-term care facilities [5]. Risk factors for CDI include the use of broad-spectrum antibiotics, which disrupt the normal gut flora, immunosuppression, recent gastrointestinal surgery, and the use of proton pump inhibitors [6]. Furthermore, the spread of hypervirulent strains, such as ribotype 027, has further exacerbated the clinical burden of CDI, leading to outbreaks and high mortality rates (around 18%) [4,7]. The treatment for CDI typically involves the cessation of the inciting antibiotic, alongside specific antimicrobial therapy directed against *C. difficile*, such as metronidazole, vancomycin, or fidaxomicin [8]. Despite advances in treatment for it, CDI remains associated with significant mortality, particularly in severe cases. Mortality rates can vary widely, with severe infections resulting in death rates as high as 20–30% [9]. In Europe, the burden of CDI is particularly high in healthcare settings, where outbreaks can lead to increased healthcare costs and significant patient mortality. Indeed, according to a recent systematic review, the highest CDI incidence in Europe was reported in Poland (6.18 per 10,000 patient days), while the lowest was found in the United Kingdom (1.99 per 10,000 patient days) [10]. This variability underscores/emphasizes the importance of stringent infection control measures and effective treatment protocols across different healthcare systems. At the same time, patients with sepsis associated with *C. difficile* are more critically ill and have significantly worse outcomes compared to those with systemic inflammatory response syndrome (SIRS) associated with the same microorganism. In this regard, when CDI was linked with SIRS or sepsis, staphylococci and enterococci species were frequently isolated from blood cultures, worsening the patient’s prognosis [11]. Inflammatory biomarkers have become invaluable tools for the management of critically ill patients, providing insights into the underlying inflammatory state and aiding in prognostication. C-reactive protein (CRP), procalcitonin (PCT), and hemogram-derived ratios have been studied in several infectious diseases, such as sepsis and Coronavirus Disease-19 (COVID-19) [12]. The latter has significantly influenced the CDI burden. Indeed, the widespread use of broad-spectrum antibiotics for treating COVID-19-associated bacterial infections has increased the risk of CDI, as these antibiotics disrupt the normal gut microbiota [13]. Furthermore, the strain on the healthcare system and the prioritization of COVID-19 patients, through the creation of appropriate wards, have led to decreased routine infection control measures. The pandemic’s impact on CDI highlights the complex interplay between infectious diseases and underscores the need for vigilant antibiotic stewardship and robust infection control practices [14]. At the same time, the role of these biomarkers in predicting outcomes, such as mortality, in CDI patients was poorly investigated. There is an urgent need to identify and validate new biomarkers specific to CDI, as current ones may not fully capture the pathogenetic processes associated with this infection. Novel investigations could potentially lead to improved risk stratification and targeted interventions for those at high risk, ultimately improving clinical outcomes in this vulnerable patient population [15]. For this reason, this study aims to evaluate the accuracy of several inflammatory biomarkers and hemogram-derived ratios in predicting mortality in patients affected by CDI.

## 2. Results

To evaluate whether biomarkers and hemogram-derived ratios can predict the mortality risk in 230 Italian (44/230) and Romanian (186/230) patients with CDI, a receiver operator characteristic (ROC) analysis was performed (Table 1). Specifically, neutrophil-to lymphocyte ratio (NLR) showed a sensitivity of 72.5%, a specificity of 58.42%, and an area under curve (AUC) = 0.652 (95% confidence interval [CI]: 0.559–0.736); systemic immune-inflammation index (SII) showed a sensitivity of 77.5%, a specificity of 54.74%, and an AUC = 0.64 (95% CI: 0.559–0.736); platelet-to-neutrophil ratio (PNR) showed a sensitivity of 2.5% and a specificity of 98.95% along with an AUC = 0.616 (95% CI: 0.514–0.711); neutrophils showed similar values for sensitivity and specificity (62.5% and 60%, respectively) along with an AUC = 0.614 (95% CI: 0.515–0.715); derived neutrophil-to lymphocyte ratio (dNLR) showed a similar sensitivity and specificity (62.5% and 62.23%, respectively) along with an AUC = 0.609 (95% CI: 0.511–0.713); lymphocytes showed a sensitivity of 5% and a specificity of 96.32% along with an AUC = 0.595 (95% CI: 0.498–0.689); CRP showed a sensitivity of 65% and a specificity of 53.68%, with an AUC = 0.594 (95% CI: 0.495–0.686); leukocytes showed a sensitivity of 62.5% and a specificity of 56.08% along with an AUC = 0.569 (95% CI: 0.461–0.672); platelets showed a sensitivity of 85%, a specificity of 33.16% along with an AUC = 0.565 (95% CI: 0.468–0.655); platelet-to-lymphocyte ratio (PLR) showed a sensitivity of 85% and a specificity of 33.16% along with an AUC = 0.558 (95% CI: 0.459–0.656); finally, PCT showed a sensitivity of 100%, a specificity of 7.41% and an AUC = 0.528 (95% CI: 0.434–0.628).

At the same time, the multivariate logistic regression model was applied to analyze mortality and adjusted for confounding factors such as age and gender. Our analysis did not show any statistically significant values for age ≥ 70 years (adjusted odds ratio; OR = 1.92, 95% CI: 0.89–4.42; *p* = 0.107), cohort (adjusted OR = 1.83, 95% CI: 0.66–5.86; *p* = 0.271), comorbidities score (adjusted OR = 1.1, 95% CI: 0.85–1.42; *p* = 0.476), and department (adjusted OR = 0.4, 95% CI: 0.13–1.02; *p* = 0.075), as shown in Table 2.

Table 3 shows the characteristics of the study population stratified according to survival. The majority of the deceased patients were Romanians (*n* = 34, 85% vs. *n* = 152, 80%; *p* = 0.465) and aged ≥ 70 years (*n* = 30, 75% vs. *n* = 108, 57%; *p* = 0.33). Additionally, these patients had a significantly higher percentage of comorbidities such as obesity (*n* = 7, 17% vs. *n* = 11, 6%; *p* = 0.021) and COVID-19 (*n* = 16, 40% vs. *n* = 39, 20%; *p* = 0.009) compared to the group of surviving patients. Conversely, most surviving patients came from a surgical department (*n* = 52, 27% vs. *n* = 5, 12%; *p* = 0.048). At the same time, the group of decease subjects showed significantly higher levels of laboratory parameters such as neutrophils [7 (5.1–9.39) vs. 8.64 (6.25–12.78) 109/L; *p* = 0.024], dNLR [4.84 (2.79–7.39) vs. 3.14 (2.08–5.36); *p* = 0.03], NLR [9 (6.21–17.3) vs. 5.64 (3.2–10.81); *p* = 0.03], and SII [2185.89 (1586.12–3335.59) vs. 1450.76 (767.1–2359.79); *p* = 0.005], but significantly lower levels of PNR [27.14 (16.71–42.47) vs. 34.24 (23.17–51); *p* = 0.022] compared to the group of surviving subjects.

A secondary aim was to assess the prevalence of CDI in different hospital settings. In this regard, our data showed a prevalence of CDI of 4.7% (44/934) among Italian patients and 15.2% (186/1224) among Romanians. Considering the prevalence by year (Figure 1), the trend was stable for the Italian cohort (around 6% from 2020 to 2022), except for 2023, where no positivity was recorded. At the same time, considering the Romanian cohort, the trend increased from 11% in 2020 to 18.3% in 2023.

Subsequently, all patients, stratified according to their cohort and their general characteristics, were evaluated, as reported in Table 4. The Romanian cohort was older compared to the Italian cohort [73 (63–81) vs. 67.5 (59.25–81.5) years; *p* = 0.306], and the Italian cohort had a higher percentage of female patients (*n* = 29, 66% vs. *n* = 94, 50%; *p* = 0.066) and urban residents (*n* = 30, 68% vs. *n* = 114, 61%; *p* = 0.266) compared to the Romanian cohort. Almost all Italian patients were admitted to a medical department (*n* = 42, 95% vs. *n* = 131, 70%; *p* < 0.001) but had a significantly shorter hospitalization period [15.5 (7–27) vs. 19.5 (14–29); *p* = 0.01]. However, they had a significantly higher percentage of comorbidities such as dyslipidemia (*n* = 13, 29% vs. *n* = 4, 2%; *p* < 0.001), hypertension (*n* = 22, 50% vs. *n* = 13, 7%; *p* < 0.001), and COVID-19 (*n* = 16, 36% vs. *n* = 39, 21%; *p* = 0.031). Despite this, the Romanian cohort showed the highest percentage of deaths (*n* = 34, 18% vs. *n* = 6, 14%; *p* = 0.465). Finally, laboratory parameters have not shown significant differences between the two groups.

## 3. Discussion

### 3.1. Potential Biomarkers, Associated Risk Factors, and Outcome

CDI is a worldwide public health problem that has been exacerbated by the overuse of antibiotics during the COVID-19 pandemic [16]. Moreover, despite numerous efforts to search for new potential biomarkers, there is a lack of reliable biomarkers that can predict mortality in CDI patients [17]. This evidence led us to investigate the potential role of several inflammatory biomarkers and hemogram-derived ratios in predicting the mortality risk in patients with CDI. According to our analysis, NLR and SII showed moderate accuracy with AUCs of 0.652 and 0.64, respectively. These hemogram-derived ratios, along with dNLR and neutrophils, demonstrated reasonably good sensitivity and specificity, suggesting their potential as tools for identifying high-risk patients. However, recent studies suggest using a combination of multiple biomarkers to predict mortality risk in these patients. Indeed, the combined application of tumor necrosis factor-α, interleukin (IL)-8, chemokine (C-C motif) ligand 5, IL-6, IL-15, and soluble suppression of tumorigenicity-2 showed high accuracy (AUC = 0.86) [18]. Similarly, the combined use of IL-8, PCT, C--X--C motif chemokine ligand 5, interferon-gamma inducible protein of 10 kDa, and IL-2Rα showed an AUC = 0.89 in predicting 30-day mortality [19]. According to Walker et al., neutrophil/white blood cell counts, PCR, and albumin are the main biomarkers associated with CDI that are highly prognostic for short-term mortality [20]. Moreover, it is important to highlight that the use of hemogram-derived ratios to predict mortality risk in the context of COVID-19 has yielded similar results to our findings, with an AUC = 0.66 and =0.65 for NLR and dNLR, respectively [21]. This outcome was confirmed by two previous studies with similar results [22,23]. Furthermore, the neutrophil-to-platelet ratio showed the highest predictive value of intensive care unit admission in COVID-19 patients (OR = 1.11, 95% CI: 0.98–1.22, *p* = 0.055) [24]. At the same time, our multivariate logistic regression model for mortality, adjusted for confounding factors such as age and gender, did not yield statistically significant results for several variables. The lack of statistical significance for age ≥ 70 years as a predictor of mortality in our study contrasts with existing literature, which often identifies advanced age as a significant risk factor for poor outcomes in CDI patients [25]. One possible explanation could be the relatively small sample size, which may limit the capacity of our analysis to detect significant associations. Similarly, the cohort and department variables did not reach statistical significance, suggesting that being part of the Italian or Romanian cohort did not predict mortality on its own. This finding might be influenced by the differences in healthcare systems, patient management protocols, and other unmeasured socio-economic factors between the two countries [26]. The comorbidities score did not show a significant association with death risk either. This result is somewhat unexpected, given that a high number of comorbidities usually correlate with poorer outcomes. However, these findings are in line with the hypothesis of the “obesity paradox”, where certain factors like obesity may play a complex role, potentially offering some protective effects against CDI [27]. Stratifying patients by their outcomes revealed that deceased patients had more comorbidities like obesity and COVID-19 compared to survivors. This contradicts some literature suggesting obesity does not worsen prognosis but supports the above-mentioned paradox [28,29]. In contrast, patients with both CDI and COVID-19 had worse outcomes than those with only COVID-19 [25]. Furthermore, deceased CDI patients also had significantly higher levels of neutrophils, dNLR, NLR, and SII and significantly lower PNR levels compared to survivors, supporting their role as potential biomarkers that can evaluate the mortality risk in CDI patients.

### 3.2. Epidemiological and Clinical Differences between the Two Cohorts

The observed prevalence of CDI was higher in the Romanian cohort (15.2%) compared to the Italian cohort (4.7%). Notably, the prevalence in Romania increased from 11% in 2020 to 18.3% in 2023, whereas the prevalence in Italy remained stable at approximately 6% until 2023, when no cases were recorded. However, it must be considered that, over the years, an increase in CDI cases took place in Italy. In fact, a recent retrospective study performed in a tertiary care hospital reported an incidence of 0.3 per 10,000 patients in 2013, rising to 5.6 in 2022 [30]. This evidence had already been reported in a previous study by the same research group, which showed an increasing prevalence from 2009 (1.7%) to 2019 (17%) in their hospital setting [31]. Regarding the Romanian setting, the estimated prevalence between 2013 and 2014 in nine hospitals in Romania was around 5%, while during 2018, 207 new cases were recorded, of which 172/207 (83%) were of nosocomial origin and 35/207 (17%) of unknown sources [32,33]. This discrepancy is certainly due to the different types of hospital settings and the varying number of critical patients undergoing antibiotic treatment [34]. It may also reflect differences in healthcare practices and infection control measures between the two countries. For this reason, the increase in CDI cases in Romania over the years underscores the need for enhanced active surveillance and prevention strategies [32]. The analysis of the two cohorts under investigation highlighted that among patients affected by CDI, the female gender was predominant in the Italian cohort, while the infection was equally distributed between males and females in the Romanian cohort, where patients were also older. Usually, the female gender is more commonly affected by CDI, especially at an advanced age [35]. Almost all Italian patients were from a medical department, although Romanian patients showed a significantly higher number of hospitalization days. These data are in line with recent literature demonstrating that in 32% of CDI cases, the hospitalization period extends beyond 15 days, particularly in patients admitted to a medical department compared to a surgical department (10% vs. 3%; *p* < 0.001) [36,37]. At the same time, Italian patients had a significantly higher prevalence of comorbidities such as dyslipidemia, hypertension, and COVID-19 compared to their Romanian counterparts. This finding can be attributed to several factors: firstly, most patients affected by CDI are affected by one (20%) to more than eight comorbidities (0.08%), complicating their clinical status and management [38]. Secondly, Italy had a prevalence of 18% of metabolic syndrome in the general population, which increases to 30% when considering the Calabria Region, where the study was performed [39,40]. Regarding COVID-19, the pandemic has significantly impacted the healthcare system, making Italy the country with the highest number of cases after China, while Romanian authorities converted emergency hospitals into COVID-19 hospitals to contain the infection [41,42].

### 3.3. Strengths and Limitations 

To our knowledge, this study is the first to evaluate hemogram-derived ratios for predicting CDI mortality, incorporating data from two tertiary care hospitals in different countries. This approach highlights the variability in clinical practices and patient characteristics, emphasizing the need for stringent infection control and antimicrobial stewardship [43,44,45,46]. However, limitations include potential biases from its retrospective design and the small sample size. Furthermore, the lack of significant findings in multivariate analysis suggests that other unmeasured factors may influence mortality. Finally, predictors were not evaluated separately for CDI and COVID-19 due to the lack of comprehensive clinical data regarding COVID-19 patients.

## 4. Materials and Methods

### 4.1. Study Design and Population

In this observational retrospective study, data were collected from the two different microbiology units in Italian and Romanian hospital settings during the COVID-19 pandemic era, spanning from 1 January 2020 to 5 May 2023, a period of time chosen due to the overuse of antibiotic therapy as a risk factor for CDI [28]. The Italian setting is a tertiary care hospital in the Calabria Region, with 225 beds, serving the health needs of 340,642 residents of the Catanzaro province (15,000 km^2^), approximately 1.8 million inhabitants of the Calabria Region, and about 13.5 million people in Southern Italy [47]. The Romanian setting is a tertiary emergency hospital in the Transylvania Region, with 1542 beds across all medical and surgical wards, serving the health needs of 688,715 residents of Cluj-Napoca province (6674 km^2^) and approximately 6.4 million inhabitants of the Transylvania Region [48]. The data were gathered using FREQUENZA v12.5.3 (available in METAFORA software v12.5.3) and ATLAS Med software v2.1, and were stored and updated in a password-protected Excel^®^ v16.67 spreadsheet.

### 4.2. Routine Diagnosis

CDI diagnosis followed international guidelines: patients with diarrhea, defined as >3 unformed stools in 24 consecutive hours or less, associated with risk factors, and a positive laboratory result for toxin A or B in stool samples, isolation of a toxin-producing strain in stools, or detection by molecular techniques of a toxin-producing strain [44]. Specifically, the toxins’ serological detection was performed using C. Diff Quik Check Complete (Techlab INC, Blacksburg, VA, USA) or VIDAS^®^ (Biomerieux, Craponne, France), while the molecular analysis was performed using FilmArray gastrointestinal panel (BioFire^®^ Diagnostics, West Warehouse, UT, USA). We collected demographic, clinical, and laboratory data at the time of CDI diagnosis. Finally, COVID-19 was diagnosed after positivity to rapid antigen or molecular tests. Inclusion criteria were (i) patients aged 18 years or older and (ii) patients with CDI. Pregnant and lactating women were excluded from the study.

### 4.3. Statistical Analysis 

Data were presented as medians with interquartile ranges or as numbers with percentages. The clinical characteristics of the study population were compared across categorized groups using different statistical tests based on the data type. A *t*-test for independent samples was employed for normally distributed quantitative variables, whereas the Wilcoxon rank-sum test was used for non-normally distributed quantitative data. For categorical data, χ2 tests and Fisher’s exact tests were utilized. Additionally, to evaluate the relationship between survival vs. deceased and certain variables, a multivariate logistic regression model was applied, adjusted for confounding factors such as age and gender. The results of the regression analysis were presented as model coefficients, 95% CI, and *p* values. ROC analysis was performed to determine the accuracy of inflammatory biomarkers and hemogram-derived ratios in predicting mortality. A *p* value < 0.05 was considered significant. The analysis was performed using R software version 4.1.2 (R Foundation for Statistical Computing).

## 5. Conclusions

Our study highlights the potential of inflammatory biomarkers and hemogram-derived ratios as prognostic tools in patients with CDI. Specifically, NLR, SII, and other parameters showed moderate accuracy in predicting mortality risk among CDI patients. These findings underscore the importance of integrating these biomarkers into clinical practice so as to improve risk stratification and targeted interventions. However, further research with larger cohorts is warranted to confirm our data and to validate new potential specific biomarkers in the field of infectious diseases.

## Figures and Tables

**Figure 1 antibiotics-13-00769-f001:**
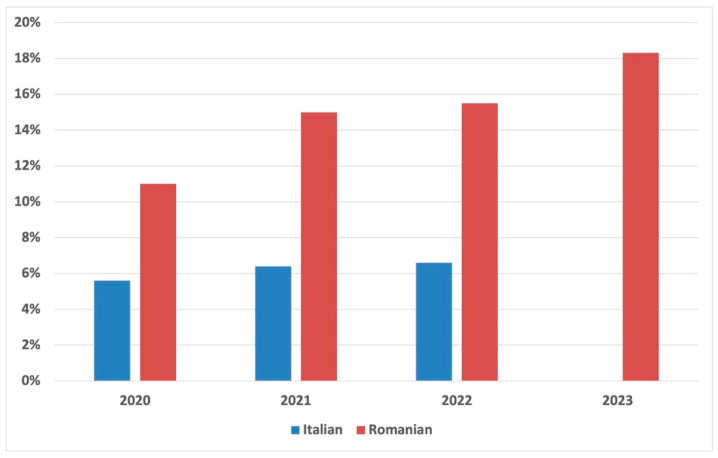
*Clostridioides difficile* infection prevalence over the years.

**Table 1 antibiotics-13-00769-t001:** ROC analysis performed on 230 patients with CDI.

Biomarker	AUC (95% CI)	Sensitivity	Specificity	Cut-Off
NLR	0.652 (0.559–0.736)	72.5	58.42	7.128713
SII	0.64 (0.543–0.729)	77.5	54.74	1568.317
PNR	0.616 (0.514–0.711)	2.5	98.95	111.2478
Neutrophils (10^9^/L)	0.614 (0.515–0.715)	62.5	60	7.48
dNLR	0.609 (0.511–0.713)	62.5	62.23	3.994118
Lymphocytes (10^9^/L)	0.595 (0.498–0.689)	5	96.32	2.37
CRP (mg/dL)	0.594 (0.495–0.686)	65	53.68	7.35
Leukocytes (10^9^/L)	0.569 (0.461–0.672)	62.5	56.08	9.44
Platelets (10^9^/L)	0.565 (0.468–0.655)	82.5	21.05	159
PLR	0.558 (0.459–0.656)	85	33.16	153.6585
PCT (ng/mL)	0.528 (0.434–0.628)	100	7.41	0.09

Abbreviations: NLR: neutrophil-to-lymphocyte ratio; SII: systemic immune-inflammation index; PNR: platelet-to-neutrophil ratio; dNLR: derived neutrophil-to-lymphocyte ratio; CRP: C-reactive protein; PLR: platelet-to-lymphocyte ratio; PCT: procalcitonin; AUC: area under curve; CI: confidence interval.

**Table 2 antibiotics-13-00769-t002:** Multivariate logistic regression model.

Variable	Adjusted OR	(95% CI)	*p* Value
Age ≥ 70 years	1.92	(0.89–4.42)	0.107
Cohort	1.83	(0.66–5.86)	0.271
Comorbidities score	1.1	(0.85–1.42)	0.476
Department	0.4	(0.13–1.02)	0.075

Abbreviations: OR: odds ratio; CI: confidence interval.

**Table 3 antibiotics-13-00769-t003:** Characteristics of the enrolled population according to survival.

	Survivors (*n* = 190)	Deceased (*n* = 40)	*p* Value
Demographic data			
Romanian cohort, *n* (%)	152 (80)	34 (85)	0.465
Age ≥ 70 years, *n* (%)	108 (57)	30 (75)	0.33
Male gender, *n* (%)	89 (47)	18 (45)	0.832
Urban area *n*, (%)	120 (63)	24 (60)	0.266
Clinical data			
Surgical area department, *n* (%)	52 (27)	5 (12)	0.048
COVID-19 ward, *n* (%)	13 (81)	8 (100)	0.526
Cancer as the main diagnosis, *n* (%)	13 (92)	4 (100)	1.00
Dyslipidemia, *n* (%)	13 (7)	4 (10)	0.506
Hypertension, *n* (%)	30 (16)	5 (12)	0.599
Obesity, *n* (%)	11 (6)	7 (17)	0.021
T2DM, *n* (%)	44 (23)	11 (27)	0.558
COVID-19, *n* (%)	39 (20)	16 (40)	0.009
Laboratory parameters, median (IQR)			
CRP (mg/dL)	6.54 (2.78–13.16)	10.66 (4.42–16.56)	0.061
PCT (ng/mL)	0.23 (0.17–0.33)	0.24 (0.18–0.35)	0.584
Leukocytes (10^9^/L)	9.01 (6.94–11.45)	9.89 (7.34–13.41)	0.17
Lymphocytes (10^9^/L)	1.23 (0.81–1.71)	1.04 (0.69–1.35)	0.06
Neutrophils (10^9^/L)	7 (5.1–9.39)	8.64 (6.25–12.78)	0.024
Platelets (10^9^/L)	235 (173–322.75)	202 (171–279)	0.199
dNLR	3.14 (2.08–5.36)	4.84 (2.79–7.39)	0.03
NLR	5.64 (3.2–10.81)	9 (6.21–17.3)	0.03
PLR	212.76 (141.57–282.02)	218.51 (164.25–307.5)	0.249
PNR	34.24 (23.17–51)	27.14 (16.71–42.47)	0.022
SII	1450.76 (767.1–2359.79)	2185.89 (1586.12–3335.59)	0.005

Abbreviations: IQR: interquartile range; T2DM: type 2 diabetes mellitus; COVID-19: Coronavirus Disease-19; CRP: C-reactive protein; PCT: procalcitonin; dNLR: derived neutrophil-to-lymphocyte ratio; NLR: neutrophil-to lymphocyte ratio; PLR: platelet-to-lymphocyte ratio; PNR: platelet-to-neutrophil ratio; SII: systemic immune-inflammation index.

**Table 4 antibiotics-13-00769-t004:** General characteristics among the Italian and Romanian cohorts of patients.

	Italian CDI (*n* = 44)	Romanian CDI (*n* = 186)	*p* Value
Demographic data			
Age (years), median (IQR)	67.5 (59.25–81.5)	73 (63–81)	0.306
Female gender, *n* (%)	29 (66)	94 (50)	0.066
Urban area *n*, (%)	30 (68)	114 (61)	0.266
Clinical data			
Medical area department, *n* (%)	42 (95)	131 (70)	<0.001
Cancer as main diagnosis, *n* (%)	5 (100)	12 (92)	1.00
Hospitalization days, median (IQR)	15.5 (7–27)	19.5 (14–29)	0.01
Hospitalization year, median (IQR)	2021 (2021–2022)	2022 (2021–2022)	0.117
Dyslipidemia, *n* (%)	13 (29)	4 (2)	<0.001
Hypertension, *n* (%)	22 (50)	13 (7)	<0.001
Obesity, *n* (%)	4 (9)	14 (7)	0.756
T2DM, *n* (%)	7 (16)	48 (26)	0.116
COVID-19, *n* (%)	16 (36)	39 (21)	0.031
Death, *n* (%)	6 (14)	34 (18)	0.465
Laboratory parameters, median (IQR)			
CRP (mg/dL)	9.01 (4.38–15.85)	6.54 (2.37–12.85)	0.053
PCT (ng/mL)	0.22 (0.11–2.38)	0.23 (0.18–0.31)	0.619
Leukocytes (10^9^/L)	9.32 (7.61–12.33)	9.14 (6.92–11.64)	0.698
Lymphocytes (10^9^/L)	1.29 (0.74–1.73)	1.17 (0.8–1.67)	0.882
Neutrophils (10^9^/L)	6.99 (5.26–10.12)	7.22 (5.18–10.15)	0.769
Platelets (10^9^/L)	233 (171–319.75)	229 (173–319)	0.568
dNLR	3.23 (2.11–5)	3.41 (2.13–5.91)	0.612
NLR	6.75 (3.16–10.45)	6.4 (3.69–11.8)	0.606
PLR	240.71 (138.16–298.17)	211.64 (146.51–281.44)	0.568
PNR	35.09 (23.82–47.95)	33.27 (21.01–50.02)	0.556
SII	1606.13 (773.78–2519.57)	1547.92 (809.88–2544.76)	0.931

Abbreviations: CDI: *Clostridioides difficile* infection; IQR: interquartile range; T2DM: type 2 diabetes mellitus; COVID-19: Coronavirus Disease-19; CRP: C-reactive protein; PCT: procalcitonin; dNLR: derived neutrophil-to- lymphocyte ratio; NLR: neutrophil-to-lymphocyte ratio; PLR: platelet-to-lymphocyte ratio; PNR: platelet-to- neutrophil ratio; SII: systemic immune-inflammation index.

## Data Availability

The original contributions presented in the study are included in the article, further inquiries can be directed to the corresponding authors.

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
