# Peer review of "Clostridioides difficile Infection: Use of Inflammatory Biomarkers and Hemogram-Derived Ratios to Predict Mortality Risk in Hospitalized Patients"

_antibiotics, 2024, doi:10.3390/antibiotics13080769_

Round 1

Reviewer 1 Report

Comments and Suggestions for Authors

The manuscript investigates the prognostic value of inflammatory biomarkers and hemogram-derived ratios in predicting mortality among patients with Clostridioides difficile infection (CDI). This retrospective study analyzed patients from two hospitals in Italy and Romania during the COVID-19 pandemic. The biomarkers evaluated include the neutrophil-to-lymphocyte ratio (NLR), systemic immune-inflammation index (SII), and platelet-to-neutrophil ratio (PNR). The study found that these markers, particularly NLR and SII, exhibited moderate accuracy in predicting mortality, with NLR showing a sensitivity of 72.5% and specificity of 58.42%, and SII a sensitivity of 77.5% and specificity of 54.74%. Other markers like PNR, neutrophils, and derived NLR demonstrated lower predictive values. The findings suggest that these biomarkers could be useful for risk stratification in clinical settings, aiding in targeted interventions for high-risk CDI patients. The limitation is that no significant biomarkers were found to predict mortality in CDI patients. Overall, this manuscript is well-written and logically sound.

Minor issues:

line 26-28: Please specify the exact time period.

line 76-78: Any supportive reference here?

Author Response

The manuscript investigates the prognostic value of inflammatory biomarkers and hemogram-derived ratios in predicting mortality among patients with Clostridioides difficile infection (CDI). This retrospective study analyzed patients from two hospitals in Italy and Romania during the COVID-19 pandemic. The biomarkers evaluated include the neutrophil-to-lymphocyte ratio (NLR), systemic immune-inflammation index (SII), and platelet-to-neutrophil ratio (PNR). The study found that these markers, particularly NLR and SII, exhibited moderate accuracy in predicting mortality, with NLR showing a sensitivity of 72.5% and specificity of 58.42%, and SII a sensitivity of 77.5% and specificity of 54.74%. Other markers like PNR, neutrophils, and derived NLR demonstrated lower predictive values. The findings suggest that these biomarkers could be useful for risk stratification in clinical settings, aiding in targeted interventions for high-risk CDI patients. The limitation is that no significant biomarkers were found to predict mortality in CDI patients. Overall, this manuscript is well-written and logically sound.

Minor issues:

  1. line 26-28: Please specify the exact time period.

Reply 1. Thank you for your comment. The text has been revised (see line 31).

  1. line 76-78: Any supportive reference here?

Reply 2. Thank you for your comment. The text has been revised (see lines 88, 372-373).

Reviewer 2 Report

Comments and Suggestions for Authors

Reviewer comment

Manuscript ID: antibiotics-3153164-peer-review-v1

Abstract:

The author should specify inflammatory biomarkers and hemogram-derived ratios in this study and report the outcome subsequently.

Lines 25-26: Our study aims to evaluate the accuracy of various inflammatory biomarkers and hemogram-derived ratios in predicting mortality among CDI patients.

Lines 31-36: NLR ... SII ... PNR, neutrophils, dNLR, and lymphocytes ... CRP, leukocytes, and platelets ... PCT... Conclusions: Neutrophils, dNLR, NLR, SII, and PNR.

The author should give a complete phrase before using its abbreviation.

How could the author differentiate this predictor for CDI or COVID-19? Some references reported The association of hemogram-derived ratios with COVID-19 mortality.

Reference:

Segalo S, Kiseljakovic E, Papic E, et al. The Role of Hemogram-derived Ratios in COVID-19 Severity Stratification in a Primary Healthcare Facility. Acta Inform Med. 2023;31(1):41-47. doi:10.5455/aim.2023.31.41-47

Asaduzzaman MD, Romel Bhuia M, Nazmul Alam Z, Zabed Jillul Bari M, Ferdousi T. Significance of hemogram-derived ratios for predicting in-hospital mortality in COVID-19: A multicenter study. Health Sci Rep. 2022;5(4):e663. Published 2022 Jun 7. doi:10.1002/hsr2.663

Velazquez S, Madurga R, Castellano JM, et al. Hemogram-derived ratios as prognostic markers of ICU admission in COVID-19. BMC Emerg Med. 2021;21(1):89. Published 2021 Jul 27. doi:10.1186/s12873-021-00480-w

Introduction

Is CDI (itself) a clinical predictor of infectious disease severity, sepsis, or mortality?

Materials and Methods

What is the design of the study? Is this a case-control or cohort retrospective? What is the inclusion and exclusion criteria?

In lines 269-281, there is no information about the study design, only data collection.

Comments on the Quality of English Language

moderate

Author Response

Abstract:

  1. The author should specify inflammatory biomarkers and hemogram-derived ratios in this study and report the outcome subsequently.

Lines 25-26: Our study aims to evaluate the accuracy of various inflammatory biomarkers and hemogram-derived ratios in predicting mortality among CDI patients.

Reply 1. Thank you for your comment. The text has been revised (see lines 25-29).

  1. Lines 31-36: NLR ... SII ... PNR, neutrophils, dNLR, and lymphocytes ... CRP, leukocytes, and platelets ... PCT... Conclusions: Neutrophils, dNLR, NLR, SII, and PNR.

The author should give a complete phrase before using its abbreviation.

Reply 2. Thank you for your comment. The text has been revised (see lines 25-29; 35).

  1. How could the author differentiate this predictor for CDI or COVID-19? Some references reported the association of hemogram-derived ratios with COVID-19 mortality.

Reference:

Segalo S, Kiseljakovic E, Papic E, et al. The Role of Hemogram-derived Ratios in COVID-19 Severity Stratification in a Primary Healthcare Facility. Acta Inform Med. 2023;31(1):41-47. doi:10.5455/aim.2023.31.41-47

Asaduzzaman MD, Romel Bhuia M, Nazmul Alam Z, Zabed Jillul Bari M, Ferdousi T. Significance of hemogram-derived ratios for predicting in-hospital mortality in COVID-19: A multicenter study. Health Sci Rep. 2022;5(4):e663. Published 2022 Jun 7. doi:10.1002/hsr2.663

Velazquez S, Madurga R, Castellano JM, et al. Hemogram-derived ratios as prognostic markers of ICU admission in COVID-19. BMC Emerg Med. 2021;21(1):89. Published 2021 Jul 27. doi:10.1186/s12873-021-00480-w

Reply 3. Thank you for your comment. The text has been revised (see lines 206-209, 277-279, 326-328).

Introduction

  1. Is CDI (itself) a clinical predictor of infectious disease severity, sepsis, or mortality?

Reply 4. Thank you for your comment. The text has been revised (see lines 75-80).

Materials and Methods

  1. What is the design of the study? Is this a case-control or cohort retrospective? What is the inclusion and exclusion criteria?

In lines 269-281, there is no information about the study design, only data collection.

Reply 5. Thank you for your comment. The text has been revised (see lines 282, 305-306).

  1. English language revision.

Reply 6. Thank you for your comment. A revision of the entire manuscript was made by an English language Editor.